# Nanofungicides with Selenium and Silicon Can Boost the Growth and Yield of Common Bean (*Phaseolus vulgaris* L.) and Control Alternaria Leaf Spot Disease

**DOI:** 10.3390/microorganisms11030728

**Published:** 2023-03-11

**Authors:** Naglaa A. Taha, Salem Hamden, Yousry A. Bayoumi, Tamer Elsakhawy, Hassan El-Ramady, Svein Ø. Solberg

**Affiliations:** 1Plant Pathology Research Institute, Agriculture Research Center, Giza 12619, Egypt; 2Agricultural Botany Department, Faculty of Agriculture, University of Kafrelsheikh, Kafr El-Sheikh 33516, Egypt; 3Horticulture Department, Faculty of Agriculture, University of Kafrelsheikh, Kafr El-Sheikh 33516, Egypt; 4Agriculture Microbiology Department, Soil, Water and Environment Research Institute (SWERI), Sakha Agricultural Research Station, Agriculture Research Center, Kafr El-Sheikh 33717, Egypt; 5Soil and Water Department, Faculty of Agriculture, University of Kafrelsheikh, Kafr El-Sheikh 33516, Egypt; 6Faculty of Applied Ecology, Agriculture and Biotechnology, Inland Norway University of Applied Sciences, 2401 Elverum, Norway

**Keywords:** biotic stress, phytopathogen, nanofungicide, antioxidants, electrolyte leakage, selenium, silica

## Abstract

There is an urgent need to reduce the intensive use of chemical fungicides due to their potential damage to human health and the environment. The current study investigated whether nano-selenium (nano-Se) and nano-silica (nano-SiO_2_) could be used against the leaf spot disease caused by *Alternaria alternata* in a common bean (*Phaseolus vulgaris* L.). The engineered Se and SiO_2_ nanoparticles were compared to a traditional fungicide and a negative control with no treatment, and experiments were repeated during two successive seasons in fields and in vitro. The in vitro study showed that 100 ppm nano-Se had an efficacy rate of 85.1% on *A. alternata* mycelial growth, followed by the combined applications (Se + SiO_2_ at half doses) with an efficacy rate of 77.8%. The field study showed that nano-Se and the combined application of nano-Se and nano-SiO_2_ significantly decreased the disease severity of *A. alternata*. There were no significant differences among nano-Se, the combined application, and the fungicide treatment (positive control). As compared to the negative control (no treatment), leaf weight increased by 38.3%, the number of leaves per plant by 25.7%, chlorophyll A by 24%, chlorophyll B by 17.5%, and total dry seed yield by 30%. In addition, nano-Se significantly increased the enzymatic capacity (i.e., CAT, POX, PPO) and antioxidant activity in the leaves. Our current study is the first to report that the selected nano-minerals are real alternatives to chemical fungicides for controlling *A. alternata* in common beans. This work suggests the potential of nanoparticles as alternatives to fungicides. Further studies are needed to better understand the mechanisms and how different nano-materials could be used against phytopathogens.

## 1. Introduction

The common bean *Phaseolus vulgaris* L. is one of the superior leguminous vegetables worldwide and is used for green pod and dry seed consumption. Brazil and Mexico are the largest producers with productions of 495,100 and 373,750 MT per year, respectively [1]. The common bean is the third most important food legume worldwide, after soybean and peanut. Total harvested area is 34.8 million ha year^−1^, which produces about 27.5 million tons year^−1^ [2]. The common bean is rich in protein (22%), dietary fiber, fat, and carbohydrates (62%), and the plant contains valuable phytochemicals and antioxidants, as well as acceptable levels of various vitamins and minerals [3]. Growing beans can improve the soil properties and reduce the nitrogen fertilization due to the plant’s characteristic of N-fixation, which increases the soil fertility [4]. However, its production faces many challenges, including pests and diseases, that are related to various bacteria [5,6], fungi [1], nematodes and insects [7], as well as abiotic stresses, e.g., salinity [8], drought [9], and heat [10]. Losses in productivity and quality have commonly reached as high as 60% due to biotic and abiotic stresses [9]. 

Fungal phytopathogens have caused severe symptoms, including leaf and pod spots, leaf blight, rust, and root-rot [11]. *Alternaria* spp. are destructive to crops such as the apple *Malus domestica* Borkh. [12], the tomato *Solanum lycopersicum* L. [13], the potato *Solanum tuberosum* L. [14], and the common bean [11]. *Alternaria alternata* caused leaf spot and leaf blight diseases [15]. Other small-spored *Alternaria* species include *A. arborescens* E.G. Simmons and *A. tenuissima* (Kunze) Wiltshire, which both caused brown leaf spots, as well as *A. solani* Sorauer, which caused early blight [16]. To control these phytopathogens, chemical fungicides have been applied [17]; however, the overuse of these chemicals may represent a real threat to the environment, as well as a serious risk to human health [18]. Although biological control strategies have been developed, they are expensive and, often, take a long time to be effective, so many farmers regard them as infeasible alternatives [19]. Consequently, research has continued for additional alternatives with low environmental and health risks [20]. One such alternative is nano-minerals [21,22,23]. 

Searching for alternatives to traditional pesticides is an emerging research topic. Much research has been focused on developing novel “sustainable pesticides”, which include nutrient-fungicides such as CuO-NPs [24]. The protective role of many elements in nano-form has been confirmed against different plant pathogens, including bacteria, fungi, actinomycetes, and nematodes [25]. These nano-metals/metalloids have included titanium (TiO_2_-NPs) [26,27], silver (Ag-NPs) [28], magnesium (MgO-NPs) [29], silicon (SiO_2_-NPs) [30], copper (Cu-NPs) [31], zinc (ZnO-NPs) [32], and other nanomaterials [33]. 

Certain nanoparticles (NPs) of Se and SiO_2_ are considered nanofungicides due to their effect against a number of phytopathogens [22,23,30]. Nanofungicides are interesting to explore due to their low dose requirement; low dose-dependent toxicity; high solubility and permeability; targeted delivery; enhanced bioavailability; and controlled release [23]. Nano-selenium and nano-silica are well-known as anti-stressors for various cultivated plants, as confirmed in many studies, and many studies have been carried out to examine their role in supporting plant cultivation under abiotic stress [34], such as drought on strawberry *Fragaria* spp. [35], the toxicity of heavy metals on rice *Oryza sativa* L. [36], and salinity on rice [37]. However, few studies have reported on the combined application of nano-Se and nano-silica to mitigate biotic stress. In one study, their combined application was used against root-rot disease induced by *Fusarium* spp. on bread wheat *Triticum aestivum* L. [38]. Furthermore, the use of nano-selenium was reported in different studies, e.g., against tomato leaf blight caused by *Alternaria alternata* [26] and against tomato late-blight disease [39]. Although nano-silicon has the potential for mitigating biotic stress in plants [40,41], more investigations are required. This includes nano-silicon’s effect against different diseases, such as stem canker; stem and leaf blight; leaf and root wilt; leaf spot; and soft rot [42]. The use of nutrients in nano-form has demonstrated their protective role against phytopathogens. Hence, it is worth exploring nanofungicides and the potential of this kind of fungicide. This could provide new directions for the application of sustainable nanofungicides as a potential replacement for traditional chemical fungicides. Therefore, this work aimed to study the role both of nano-selenium (nano-Se) and nano-silica (nano-SiO_2_), both individually and combined, in controlling leaf spot disease caused by *Alternaia alternata* in the common bean. Nano-Se and nano-silica were compared to a commercial fungicide and a negative control with no treatment, during two successive seasons. In addition to yield measurements, the chlorophyll content, the enzymatic activities, and the total antioxidants in plant leaves were studied after the applications.

## 2. Materials and Methods

### 2.1. Pathogen Isolation, Purification, and Pathogenicity Tests 

Six pathogenic fungi were isolated from the infected leaves of common bean plants, showing typical leaf spot symptoms, that were obtained from commercial fields in El Beheira Governorate, Egypt. Infected leaves were washed, cut into small pieces (5 mm), surface sterilized with a sodium hypochlorite solution (0.5%) for 2–3 min, and then washed 3 times using sterilized water. Samples were dried between 2 layers of sterilized filter papers and moved to a potato dextrose agar (PDA) medium in 9 cm Petri dishes at 28 ± 2 °C and cultivated for 72 h. Pure cultures were collected for each of the six isolates using the hyphal tip technique. The purified isolates were confirmed for pathogenicity on a susceptible cultivar Giza 12, which was obtained from the Horticulture Research Institute (Agricultural Research Center, Sakha, Egypt).

The plants were grown in pots 30 cm in diameter with 4 plants per pot, and they placed in a greenhouse. The number of replicates for each pathogen isolate was five. All fungal isolates were prepared using conidial suspensions in distilled water at a rate of 5 × 10^5^ spores mL^−1^ taken from 10-day-old PDA cultures. The spore suspensions were sprayed on the whole bean plants at 35 days old. Based on the protocol established by Panwar et al. [43], the symptoms were observed two weeks after inoculation, and the disease severity was recorded. The most virulent pathogenic isolate was selected for further in vitro experiments. The most virulent isolate was identified as *Alternaria alternata* based on their morphological features and microscopic parameters, as described by Ozcelik and Ozcelik [44] in the Mycology and Disease Survey Research Department (Plant Pathology Research Institute, ARC, Giza, Egypt). 

### 2.2. Preparing Nanoparticles

Preparing Se-nanoparticles was achieved using isolated microbes, which were obtained from soil samples collected from the experimental farm at the Sakha Agricultural Research Station (Kafr El-Sheikh, Egypt). These isolates were mainly selected based on their potential to tolerate high concentrations of selenium (as a selenite). These isolates were also screened for selenite tolerance using tryptone soy broth (TSB), which was amended with Na_2_SeO_3_ (300 mg L^−1^). Nutrient broth medium was prepared, and a sterilized sodium hydrogen selenite (NaHSeO_3_) solution was supplemented from a 10,000 mg L^−1^ stock solution to reach 300 mg L^−1^ concentration. The isolated microbes were identified as *Bacillus cereus*, and the TAH strain of this species was used for the biosynthesis of nano-selenium. The biologically synthesized selenium ranged 41–102 nm in size and were produced at the Agricultural Microbiology Research Department (SWERI, ARC, Giza, Egypt), according to Ghazi et al. [45]. The high-resolution transmission electron microscopy was used to measure the size of the nanoparticles (HR-TEM, Tecnai G20, FEI, The Netherlands), and this was conducted by the Nanotechnology and Advanced Material Central Laboratory, ARC. Silicon oxide nanoparticles, NPs-SiO_2_, were prepared by fine-grinding and purchased from Agricultural Microbiology Laboratory (SWERI, ARC, Giza, Egypt) and with a diameter of 10 nm, a specific surface area of 260–320 m^2^ g^−1^, and a pH of 4–4.5.

### 2.3. In Vitro Experiments

Different doses of nano-selenium (nano-Se) and nano-silica (nano-SiO_2_), alone and together, were tested against the most virulent isolate (A5) of *A. alternata.* This was conducted in vitro. The applied doses of nano-Se were 25, 50, and 100 ppm, whereas nano-SiO_2_ were 100 and 200 ppm. The combined treatment had half of the highest doses of each compound (i.e., 50 ppm nano-Se + 100 ppm nano-SiO_2_). All applied doses of the nanoparticles were added to 100 mL of PDA media. Each dose had 5 replicates in different Petri dishes (each 9 cm in diameter). When the fungal cultures were 7 days old, 5 mm agar plugs were obtained from the edges (with vigorously growing fungi) and inoculated at the middle of the plates with PDA media, plus the different rates of nanoparticles; this was performed on all five replicates. The plates were incubated for 10 days at 28 ± 2 °C. Under full growth, the colony diameters were measured and then compared to the control plates. According to Ferreira et al. [46], the inhibiting growth percents were calculated and compared to the negative control (no treatment) using the following formula:$$Reduction\;rate\;\left(\%\right)=\frac{R-r}{R}\times 100$$
where (*R*) is the radial growth of fungi as a control and (*r*) is the radial growth of fungi in the treated plates.

### 2.4. Field Experiments

The field experiments were carried out in the common bean cv. Giza 12, which had been grown under open field conditions during the summer seasons 2021 and 2022 at a private farm in Aljazeera village, El Rahmaniya city, El Beheira Governorate, Egypt (31.1°06′19.4″ N 30.6°37′47.9″ E). This farm was chosen due to its disease history, as *Alternaria alternata* has regularly been causing problems, and its frequent bean cultivation. A randomized complete block design was arranged with four replicates. Each plot was 12 m^2^ (4 m long and with 5 rows each 0.6 m broad). Bean seeds were spaced 10 cm on ridges, and the sowing was carried out on the 20th of February and 5th of March in 2021 and 2022, respectively. An overview of the treatments is provided in Table 1. 

The severity degree of the leaf spot disease was rated in the field, according to Panwar et al. [43]: 0 = no symptoms of leaf spot; 1 = up to 1% indication of disease in leaf area; 3 = 1–10% indication of disease in leaf area; 5 = 11–25% indication of disease in leaf area; 7 = 26–50% indication of disease in leaf area; and 9 = >50% indication of disease in leaf area. The scoring was conducted 45, 60, 75, and 90 days, from sowing and the disease severity was calculated using the following formula: Disease severity (%) = [Sum of all disease rating/(Total number of ratings × Maximum disease grade)] × 100.
Efficacy (E%) of treatments against the pathogen was calculated as: E% = [(A − B)/A] × 100,
where E (efficacy percent), A (disease severity of control), and B (disease severity of treatment).

The mean of the area under the disease progress curve (AUDPC) for each replicate was assessed according to Pandy et al. [47]: AUDPC = D [1/2 (Y1 + Yk) + (Y2 + Y3 + …… + Yk − 1)], where D (time interval), Y1 (first disease severity), Yk (last disease severity), Y2, Y3, … and Yk − 1 (intermediate disease severity).

### 2.5. Vegetative Growth Traits, Yield and Photosynthetic Attributes

Vegetative growth traits, including the number of leaves per plant, stem length, fresh and dry weight of common bean plants, were measured during both seasons. All vegetative growth traits were measured 70 days from sowing during both seasons. Chlorophyll a and b contents (mg 100 g^−1^ FW) were measured in plant leaves using the method of Nagata and Yamashita [48] by spectrophotometer analysis at the wavelengths 645 and 663 nm and calculated by substituting the readings in the following equations: Chl. a = 0.999 × A_663_ − 0.0989 × A_645_
Chl. b = −0.328 × A_663_ + 1.77 × A_645_

Chlorophyll fluorescence was measured as an important parameter for photosynthetic performance and its response to stress. This parameter expressed the maximum efficiency of the photosystem PSII (Fv/Fm) and was measured using a portable Optic-Science OS-30p + Fluorometer (Opti-Sciences, Inc., Hudson, NH, USA), according to Maxwell and Johnson [49]. 

Plants were harvested at the seed maturity stage. Dry seeds were manually extracted and recorded as the weight of seeds per plant (g) and per plot (g). Thereafter, the value was calculated to provide the seed yield in Mg (tons) ha^−1^. 

### 2.6. Antioxidant Enzymes, Antioxidative Activity and Electrolyte Leakage

To determine the antioxidant enzymatic activities (catalase, peroxidase, and polyphenol oxidase), 0.5 g of fully expanded young leaves were homogenized in liquid nitrogen with 3 mL of extraction buffer (50 mM TRIS buffer (pH 7.8) containing 1 mM EDTA-Na_2_ and 7.5% polyvinylpyrrolidone)) using a pre-chilled mortar and pestle. The homogenate was filtered through 4 layers of cheesecloth and centrifuged at 12,000 rpm for 20 min at 4 °C. The supernatant, which was re-centrifuged at 12,000 rpm for 20 min at 4 °C, was then used for the total soluble enzymatic activity assay. The enzymatic activities were measured colorimetrically using a double-beam UV/visible spectrophotometer Libra S80PC (TechnoScientific Company, Nottingham, UK), 70 days from sowing.

Catalase (CAT; EC 1.11.1.6) activity was measured by following the consumption of H_2_O_2_ at 240 nm, according to Aebi [50]. A total of 1 mL of the reaction mixture contained 20 mg total protein, 50 mM sodium phosphate buffer (pH 7.0), and 10 mM H_2_O_2_. The reaction was initiated by adding the protein extract. For each measurement, a blank corresponded to the absorbance of the mixture at time zero, and the actual reading corresponded to the absorbance after 1 min. One unit of CAT activity was defined as a 0.01 decrease in absorbance at 240 nm mg^−1^ of protein min^−1^. 

Peroxidase (POX; EC 1.11.1.7) activity was determined according to the procedure proposed by Rathmell and Sequeira [51]. The reaction mixture consisted of 2.9 mL of a 100-mM sodium phosphate buffer (pH 6.0 containing 0.25% (*v/v*) guaiacol (2-methoxy phenol) and 100 mM H_2_O_2_). The reaction was started by adding 100 mL of crude enzyme extract. Changes in absorbance at 470 nm were recorded at 30 s intervals for 3 min. Enzymatic activity was expressed as an increase in the absorbance min^−1^·g^−1^ fresh weight. 

Polyphenol oxidase (PPO; EC 1.10.3.1) activity was determined according to the method described by Malik and Singh [52]. The reaction mixture contained 3.0 mL of buffered catechol solution (0.01 M) freshly prepared in 0.1 M phosphate buffer (pH 6.0). The reaction was initiated by adding 100 mL of the crude enzyme extract. Changes in the absorbance at 495 nm were recorded at 30 s for 3 min. Enzymatic activity was expressed as an increase in the absorbance min^−1^·g^−1^ fresh weight. 

Antioxidative activity in the plant tissue was determined by a DPPH (2,2-diphenyl-1-picryl hydrazyl) assay, as described by Binsan et al. [53]. In brief, plant samples were extracted, and 1.5 mL was added with 1.5 mL of 0.15 mM DPPH in 95% ethanol. The mixture was stored in the dark for 30 min at room temperature. Using a double-beam UV/visible spectrophotometer Libra S80PC (TechnoScientific Company, Nottingham, UK), the absorbance of the resulting solution was measured at 517 nm. The calibration curve was prepared using Trolox in the range of 12.5 to 100 μM. 

Electrolyte leakage was measured using an electrical conductivity meter, according to the method of Whitlow et al. [54] and later modified by Szalai et al. [55]. Twenty leaf discs (1 cm^2^) were placed individually into flasks containing 25 mL of deionized water (Milli-Q 50, Millipore, Bedford, MA, USA). Flasks were shaken for 20 h at an ambient temperature to facilitate electrolyte leakage from injured tissues. Initial EC measurements were recorded for each vial using an EC meter (Fistreem Jenway Bench Top EC meter, Model 3510 Medica Scientific Company, Stockport, UK). Flasks were then immersed in a hot water bath (Fisher Isotemp, Indiana, PA, USA) at 80 °C (176 F) for 1 h to induce cell rupture. The vials were again placed on the Innova 2100 platform shaker for 20 h at 21 °C (70 F). Final conductivity was measured for each flask. The percentage of electrolyte leakage for each sample was calculated as the initial conductivity/final conductivity × 100. Chlorophyll florescence, chlorophyll contents, all vegetative growth parameters, antioxidant enzymes, and electrolyte leakage were assessed 70 days from sowing in both growing seasons. The general layout of the entire work is overviewed in Figure 1.

### 2.7. Microscopic Examination

The fungal culture grown on the PDA media supplemented with selenium nano-particles was examined by light microscope with a magnification power of 200× by sticky tape touch method after incubation for 7 days. Samples from the control and nano-Se were also examined 5 days from the inoculation on the PDA media with a scanning electron microscope (Model: SEM, JEOL JSM 6510 Iv, Tokyo, Japan) by using the high vacuum mode at the Nanotechnology Institute (Kafrelsheikh University, Kafr El-Sheikh, Egypt). This was conducted to visualize any potential effects of the nano-Se application on *A. alternata*.

### 2.8. Statistical Analyses

The two field experiments were arranged as randomized complete block designs with four replicates for each treatment in each trial. Statistical analyses were conducted with the CoStat package program (Computer Program Analysis, Version 6.303; CoHort Software, Berkeley, CA, USA) using ANOVA and, thereafter, Duncan’s multiple range tests at 5% level of probability to compare the means [56].

## 3. Results

### 3.1. Pathogenicity Tests 

A greenhouse experiment was conducted to assess the pathogenic ability of the six isolates. The Alternaria isolates were able to infect the susceptible bean cultivar Giza 12 and caused typical leaf spot symptoms, whereas the isolate A5 (*Alternaria alternata*) was the most aggressive one in the experiment (Table 2 and Figure 2). The A5 isolate had the highest disease severity value (78.23%); however, the other isolates varied in their degrees of disease severity. Consequently, the A5 isolate of *A. alternata* was chosen for following studies. 

### 3.2. In Vitro Antifungal Activity of Nano-Se and Nano-SiO_2_

The antifungal effects of nano-Se and nano-SiO_2_ alone or in combination were examined in vitro. This was conducted on the most aggressive isolate A5. The highest antagonistic effect was found with the chemical control (95.55% reduction), followed by nano-Se at the highest dose (100 ppm), and the combined application of the nanoparticles (85.1% and 77.78% reduction, respectively). All applied nanoparticles significantly reduced the mycelial growth of the pathogen, as compared to the untreated control (Table 3; Figure 3). 

### 3.3. Microscopic Investigation

Light microscope images of the Alternaria conidia from the in vitro experiment showed an obvious effect of the nano-Se treatment. A reduction in the number and homogeneity of the conidia, in addition to an occurrence of distortions, and a clear decrease in the conidia size was observed after the nano-Se was compared to the negative controls with no treatment (Figure 4). The change in mycelial growth was examined 5 days after inoculation using a scanning electron microscope apparatus (Figure 5). The nano-Se application (100 ppm) caused morphological alterations in the hyphae and mycelial growth. The mycelium and hyphae of the *A. alternata* fungus was strictly injured in the presence of nano-Se (Figure 5A), as compared to the negative control that was untreated and showed typical mycelial structures for the fungus (Figure 5B). 

### 3.4. Disease Severity and Efficacy Percent under Field Conditions

The mean of the area under the disease progress curve (AUDPC) was significantly influenced by the different treatments (Figure 6). When left untreated (negative control), it resulted in the highest AUDPC values, followed by the nano-SiO_2_ application, as compared to the other treatments, while the nano-Se application alone or in combination with nano-SiO_2_ showed the lowest AUDPC values, and this was shown in both seasons. There were no significant differences between nano-Si and the commercial fungicide, and this was found in both seasons (Table 4). The progression of the Alternaria leaf spot disease severity was recorded 4 times: 45, 60, 75, and 90 days after sowing. The disease severity (%) was significantly influenced by the different nano-Se and nano-Si applications at all sampling dates during both seasons. The highest disease severity values were observed with untreated plants (negative control treatment), which were recorded at 29.5%, 42.1%, 61.8%, and 89.6% in the first season and at 31.3%, 49.1%, 66.0% and 94.7% in the second season at the aforementioned days, respectively. Comparatively, the commercial fungicide application (positive control) had a superior effect on the disease severity on all the dates during the growing seasons and showed the lowest disease severity percentages in all cases (Figure 7), as compared to the nanoparticles. Nano-Se alone had a better effect on disease severity (%), as compared to nano-SiO_2_ alone or in a combined reduced dose.

The efficacy percentages of all treatments were significantly impacted by the severity percentage of the Alternaria leaf spot disease, as shown in Table 4. The highest efficacy percentage was found from the fungicide treatment followed by nano-Se application, but without significant differences between them. The nano-Si application had lower values than the combined application of nano-Se and Si, which showed intermediate values in both seasons. The AUDPC was significantly influenced by the treatments (Figure 7). The untreated (negative control) group resulted in the highest AUDPC values, followed by the nano-Si application in both seasons, while nano-Se alone or in combination with nano-Si showed the lowest AUDPC values, and the commercial fungicide application showed no significant differences in its results across both seasons.

### 3.5. Vegetative Growth and Photosynthetic Traits

Under field conditions, vegetative growth traits, chlorophyll fluorescence and chlorophyll contents were examined 70 days from sowing (Table 5). The stem length, the number of leaves, and the plant fresh and dry masses were significantly enhanced by the different nanoparticle applications, as compared to the fungicide treatment (positive control) and the untreated plants (negative control). The same results were observed in both seasons. Nano-Se produced the tallest plants with the highest number of leaves per plant, which resulted in the highest fresh and dry plant biomass production. The negative controls had the lowest values in both seasons. 

In most cases, the combined application of nano-Se and nano-SiO_2_ resulted in lower values than nano-Se applied alone. The chlorophyll fluorescence (F_V_/F_M_) values were significantly increased by all nanoparticles (alone or in combination), as compared to the controls. These were followed by the fungicide treatment, while untreated plants produced the lowest values. The highest values of chlorophyll a and b were obtained by applying nano-Se and fungicide treatments, with no statistical differences in these two, while the negative control plants without any treatment produced the lowest values in both seasons. The combined application of nanoparticles and nano-Si alone had intermediate chlorophyll values.

### 3.6. Enzymatic Activities

Catalase, peroxidase, and polyphenol oxidase were assessed in common bean leaves 70 days after sowing (Figure 8). Overall, the highest values were found after applying nano-Se, followed by fungicide and combined nano-Se and nano-SiO_2_. The lowest values were found in the negative control (no treatment). 

### 3.7. Antioxidant Activity and Electrolyte Leakage

The application of nano-Se and nano-Si significantly increased the antioxidant activity, as compared to the negative control (no treatment), which resulted the lowest values (27.77 and 31.19 µM 100 g^−1^) in both seasons, respectively (Figure 9). Nano-Se application alone produced the highest antioxidant values (84.45 and 77.79 µM 100 g^−1^ in the two seasons) followed by the combined application of nano-Se/nano-Si, fungicide treatment, and finally nano-Si spraying alone in both seasons. In contrast, untreated plants had the highest electrolyte leakage percentage (90.66% and 88.48% in the two seasons). All nano- and fungicide treatments resulted in low leakage percentages, as compared to the negative control. Generally, as confirmed by the previous measuring enzymes, the best (lowest) values of electrolyte leakage and the total antioxidants were obtained after applying nano-Se. 

### 3.8. Response of Total Yield of Dry Seeds to Applied Nanoparticles

The application of nano-Se and nano-SiO_2_ alone or in combination significantly increased the dry seed yield for both growing seasons (Table 6). The highest yield was obtained from plants treated with 100 ppm nano-Se (54.22, 45.75 g/plant and 3.07, 2.66 Mg ha^−1^), followed by the combined application of both nanoparticles at a half dose (49.13, 40.18 g/plant and 2.77, 2.49 Mg ha^−1^) in both seasons. The untreated plants showed the lowest total seed yield (30.85, 27.19 g/plant and 2.17, 1.77 Mg ha^−1^) in both seasons. The intermediate values of the total dry seed yield were recorded by the application of fungicide and the 200 ppm nano-SiO_2_ treatments in both seasons.

## 4. Discussion

Agriculture faces several challenges that, in the future, may decrease crop production. This includes the pollution from the intensive use of agrochemicals and biotic stresses from pests and diseases. Therefore, the current study was carried out to examine whether nano-sized particles of selenium and silicon could be used to boost plant health and control leaf spot disease caused by the fungus *Alternaria alternata*. Therefore, we sought to answer the following questions: To what extent could nano-Se and nano-Si work against this disease? Could the applied nanoparticles of Se and Si be effective for inhibiting the fugus, as compared to the traditional fungicides? Which nano-treatments would be the most effective? What was the impact of the applied nanofungicides on common bean attributes? 

The present study involved two kinds of potential nanofungicides; the first was a biological nano-Se, and the second was nano-SiO_2_ that was prepared by a physical method. Our study was conducted to identify the most effective agent against the studied pathogen (*A. alternata*). We selected the phytopathogen of *Alternaria* in our current study. Several species of the *Alternaria* genus are well-known as phytopathogens, which infect many genera of vegetables and fruits. These species have a saprophytic nature, and their pathogenicity actions were due to toxins and cell-wall degrading enzymes [57]. Among those are *Alternaria dauci, A. solani, A. tenuis, A. brassicae*, *A. brassicicola*, and *A. alternata*. This study focused on *A. alternata* as a common phytopathogen, which could cause severe leaf spot problems in the common bean. *A. alternata* is also considered one of the most important diseases affecting common bean plants under the temperate conditions, as a serious foliar ailment causing massive yield reductions [11,26]. In addition, other crops have been infected, such as faba bean *Vicia faba* L. [15], tomato [57], apple [12], and orange *Citrus* spp. [58]. 

To what extent could the studied nano-Se and nano-SiO_2_ boost the growth and yield of the common bean as well as be used against Alternaria leaf spot disease? From our findings in the current study, it was obvious that the nano-Se and, to some extent, the nano-SiO_2_ were effective against this pathogen. The applied in vitro nano-Se alone and/or in combination with nano-SiO_2_ recorded an 85.1% and 77.8% reduction in mycelial growth, respectively. This was in line with El-Gazzar and Ismail [26], who reported 81.1% and 89.6% reduction in mycelial growth of *Alternaria* isolated from tomato when using 50 and 100 ppm nano-Se. Therefore, the inhibitory effect could be explained by the congregation and the installation of the particles. The factors of importance included the applied doses, the sizes and shapes of the particles, and the solubility and agglomeration statuses of the particles [59,60]. The results in the current study agreed with the work published by El-Gazzar and Ismail [26], who proved that 100 ppm nano-Se was effective against the leaf blight in tomato plants. Other studies have shown that nano-SiO_2_ at 200 ppm was effective against a disease complex in beetroot *Beta vulgaris* L. [32] and on early blight in tomato plants [30]. We included field trials with the examination of various growth parameters, including vegetative traits and yield. All parameters were significantly increased with nano-Se and nano-silica, as compared to the controls. The spraying of nano-Se and/or -SiO_2_ alleviated the negative impacts of the fungus stress. 

Nano-SiO_2_ and/or nano-Se have been reported to increase yields due to their ability to increase the photosynthetic and antioxidative enzymatic activities, as well as their ability to regulate the negative effects of stress, as reported in different species [34]. *A. alternata* could reduce the photosynthesis by inhibiting photosynthetic activity through the necrosis of plant tissues and by damaging structural and stomatal closures of the plants. Our study showed that the photosynthetic attributes, including the chlorophyll content (a and b) and the chlorophyll fluorescence, were significantly influenced by the stress caused by the fungus. This was shown in the positive control, whereas these attributes were less affected by applying nano-Se and/or nano-SiO_2_ than by applying a chemical fungicide. Under similar stress conditions, nano-treatments have appeared to promote the absorption and conversion of light, which enabled the treated plants to maintain normal levels of carbon assimilation [61]. The selected nanoparticles showed a significant ability to support photosynthetic attributes under biotic/abiotic stress, as reported in many published studies on other crops (e.g., [32,62,63,64]). Both nano-Si and nano-Se could increase chlorophyll content under stressful conditions and protect chloroplasts from oxidative damage, potentially because they may act as a cofactor in enzymatic reactions involved in various biosynthetic pathways [35]. 

Under biotic stresses, plants should be armed with enzymatic and/or non-enzymatic antioxidants. In our study, this included enzymatic activities, electrolyte leakage, and total antioxidants. This defense system, in general, evolves different systemic signaling pathways (i.e., calcium, reactive oxygen species (ROS), and phytohormones). The application of the studied NPs showed that they had the ability to enhance the plant resistance to phytopathogens, probably as a result of producing ROS or phytohormones [64]. We showed that the biological nano-Se recorded the highest values in CAT, PPO, and POX enzymes, as well as the total antioxidant content, followed by the combined application of nano-Se and nano-silica at half doses. This finding confirmed the protective role of both nano-Se and nano-Si against oxidative stress, in this study, as a result of biotic stress. This was shown by the increased activity of the enzymatic antioxidants, which reduced the ROS and decreased the lipid peroxidation [35]. The distinguished role of biological nano-Se was confirmed in previous studies in other crops. The effectiveness of biological nano-Se has been confirmed in different studies and under different stresses, such as cucumber under salinity and heat stress [65], wheat under Fusarium root-rot disease [66], pak choi under heavy metal toxicity [67], and rapeseed under salinity stress [68], whereas the biological silicon-NP improved the common bean yield under heavy metals and salinity stress [38]. In this study, we confirmed its effect against the Alternaria leaf spot disease in the common bean.

Nano-Se and nano-SiO_2_ are well-known as anti-stressors under biotic stress (i.e., fungal phytopathogen of *A. alternata*). This antifungal activity was confirmed as inducing plant physiological immunity against phytopathogens such as leaf blight in tomato [26] and early blight disease in eggplant *Solanum melongena* L. [69]. It was reported that spraying 200 ppm SiO_2_-NP on tomato plants under bacterial and fungal attacks promoted plant growth parameters, proline, chlorophyll, carotenoid, and enzymatic activities, and the spraying reduced the disease indices [30]. In our study, the biological nano-Se recorded the highest values in most studied parameters, including vegetative attributes, photosynthetic items, enzymatic antioxidants, and dry seed yield, as compared to the controls and nano-SiO_2_. 

Few recent publications have discussed the combined application of Se and Si (e.g., [70]), whereas the interaction between nano-nutrients such as Se and Si still needs more investigation (e.g., [19,71]). The obtained results indicated that the biological nano-Se had the best results among all treatments, whereas nano-SiO_2_ had the lowest results, ranking lower than the combined nano-Se and nano-silica. The biological nano-Se had unique physiochemical characteristics in the plant system [68], which has attracted considerable concern worldwide due to their significant potential for alleviating plant stress [63]. Nano-silica has been effective in reducing serious insect and mechanical damage. Based on the combined application of nano-Se and nano-silica, the effectiveness of nano-Se could have decreased due to a hetero-aggregation: This phenomenon can occur between nanoparticles from different sources, wherein their size increases and then interferes with the needed penetrating power.

## 5. Conclusions

Through the use of nanotechnology, the production of nano-pesticides with protective effects from elements such as Ag, Cu, Se, Si, TiO_2_, and Zn, has the potential to work against several phytopathogens. The current study is the first report on applying bio-nano-Se and nano-silica against the fungus of *A. alternata* in the common bean. The current study showed that biological nano-Se could influence the growth of the common bean positively. This was shown by the increased photosynthetic activity, better antioxidant enzyme efficiencies (CAT, PPO, and POX), and thus, better protection of the plants against this fungus, which also improved the plants’ growth and yield. The application of the selected nano-Se and nano-SiO_2_ generated beneficial effects by supporting the plant enzymatic systems and decreasing the disease severity of *A. alternata*. Although our findings may contribute to a better understanding of the antifungal effects of the applied nanofungicides, this novel antifungal alternative to chemical control requires additional investigation, particularly related to the interactions between the applied nano-Se and the nano-SiO_2._

## Figures and Tables

**Figure 1 microorganisms-11-00728-f001:**
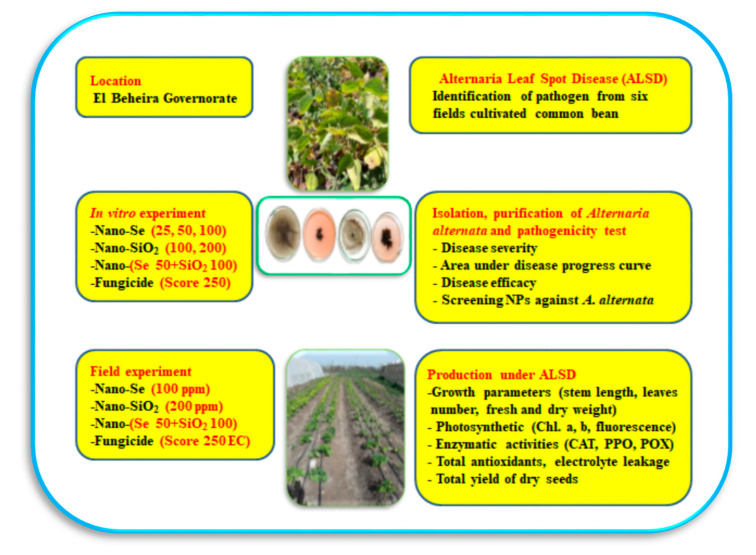
A general overview on the layout of the current study including the location, different experiments, and different measurements. Abbreviations: ALSD: Alternaria leaf spot disease; Chl.: chlorophyll; CAT: catalase; PPO: polyphenol oxidase; POX: peroxidase.

**Figure 2 microorganisms-11-00728-f002:**
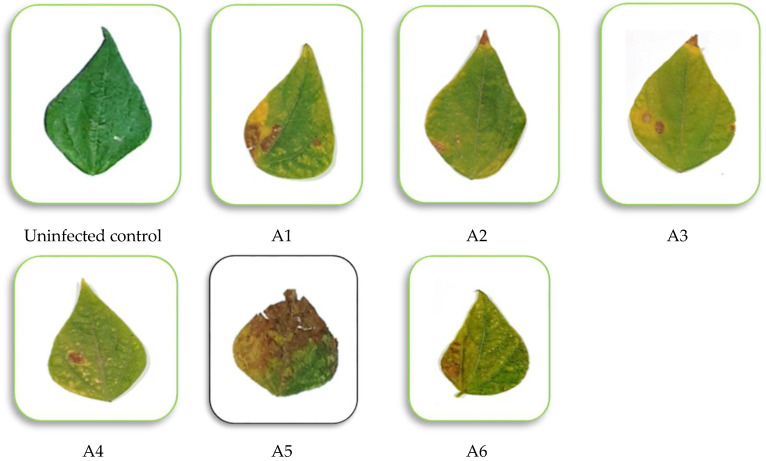
Images illustrating how the six isolates of *Alternaria alternata* affected the common bean plants. The images were taken 21 days after inoculation under greenhouse conditions.

**Figure 3 microorganisms-11-00728-f003:**
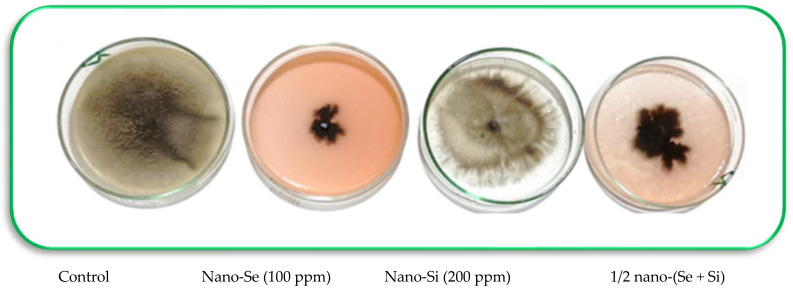
Images illustrating of the inhibition of *Alternaria alternata* mycelial growth after some of the treatments in the in vitro experiments. Control = no treatment (negative control).

**Figure 4 microorganisms-11-00728-f004:**
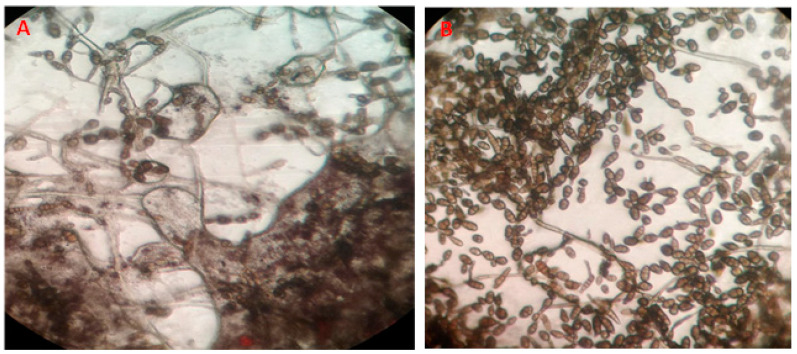
Light microscopic image (200×) of conidia of *Alternaria alternata* grown on PDA for 7 days after incubation, where (**A**) is influenced by Nano-Se (100 ppm) and (**B**) is the negative control with no treatment.

**Figure 5 microorganisms-11-00728-f005:**
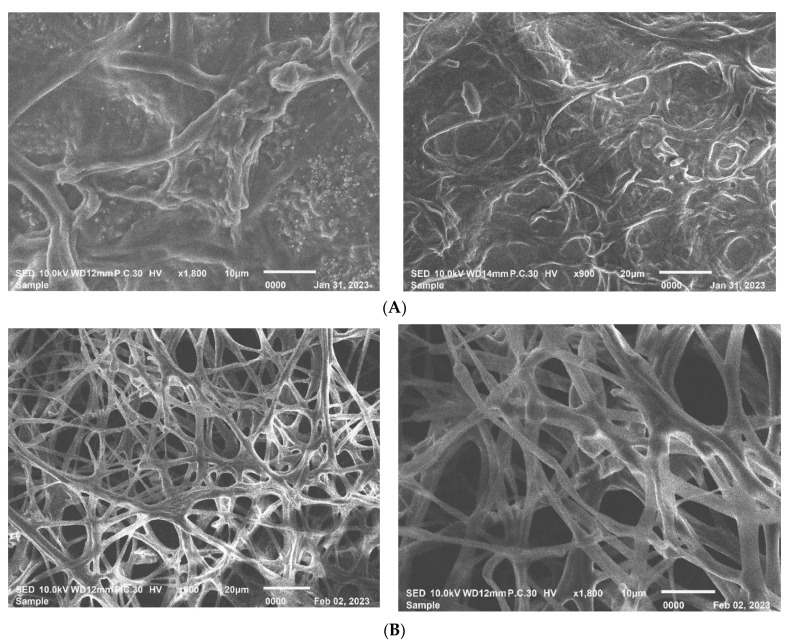
Scanning electron microscope (SEM) images of mycel of *Alternaria alternata* grown on PDA for 7 days after incubation, where (**A**) is influenced by Nano-Se (100 ppm) and (**B**) is the negative control with no treatment. (**A**) Mycelial growth of *Alternaria alternata* influenced by nano-Se application at 100 ppm. (**B**) Normal mycelial growth of *Alternaria alternata* without nano-Se application (Control).

**Figure 6 microorganisms-11-00728-f006:**
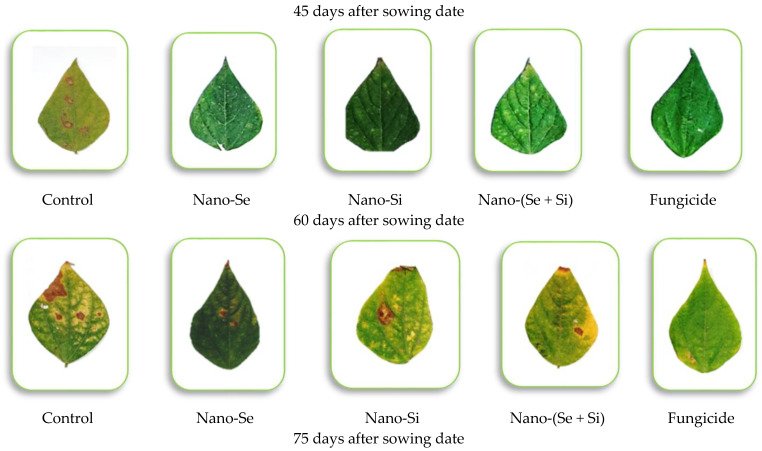
Effects of the various treatments on the development of leaf spot disease caused by *A. alternata* on common bean plants (Giza 12, cv.) grown in the field and sampled 45, 60, 75, and 90 days after sowing, and where Control = no treatment, Nano-Se = 100 ppm nano-Se, Nano-Si = 200 ppm nano-Si, Nano-(Si + Si) = combined 1/2 dose of nano-Se + nano-Si, and Fungicide = Score 250 EC at 1 mL/2 L.

**Figure 7 microorganisms-11-00728-f007:**
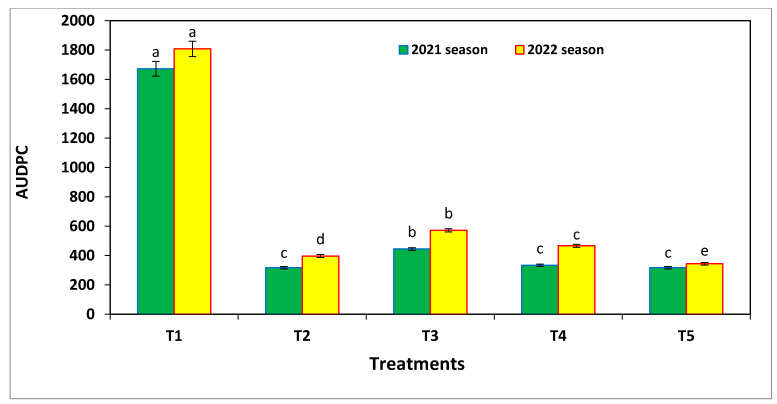
Area under disease progress curve (AUDPC) of the leaf spot disease on common bean in the field experiments, and where T1 = negative control, T2 = 100 ppm nano-Se, T3 = 200 ppm nano-Si, T4 = 1/2 nano-Se + Si, and T5 = fungicide (Score 250 EC at 1 mL/2 L). Results from the 2021 and 2022 seasons. Same letter above the error bar indicates no significant differences.

**Figure 8 microorganisms-11-00728-f008:**
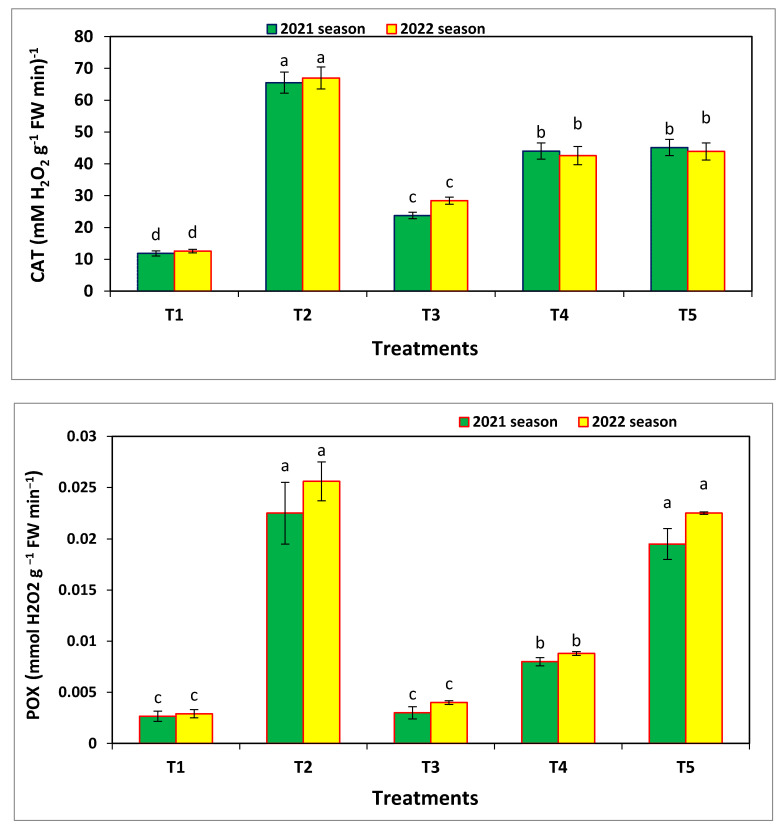
Effects of the different treatments on catalase, peroxidases, and polyphenol oxidase activity in the bean plants measured 70 days after sowing in the field experiments, and where T1 = negative control, T2 = 100 ppm nano-Se, T3 = 200 ppm nano-Si, T4 = 1/2 nano-Se + Si, and T5 = fungicide (Score 250 EC at 1 mL/2 L). Same letter above the error bar indicates no significant differences.

**Figure 9 microorganisms-11-00728-f009:**
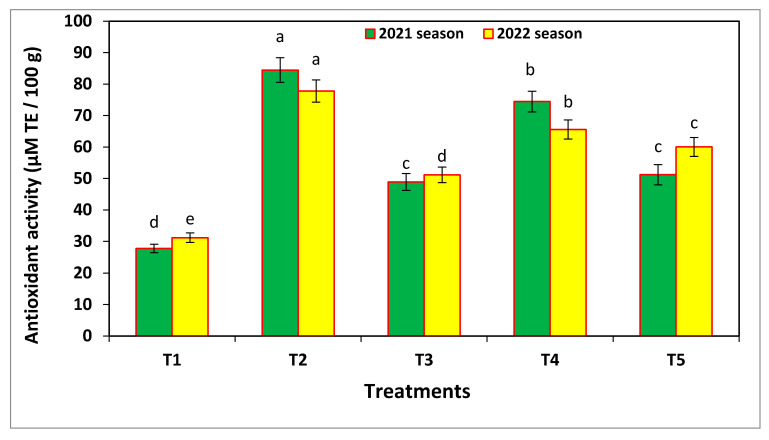
Effects of the different treatments on total antioxidant activity and electrolyte leakage measured 70 days after sowing in the field experiments in 2021 and 2022. Results are given as mean values ± standard error bars and where T1 = negative control, T2 = 100 ppm nano-Se, T3 = 200 ppm nano-Si, T4 = 1/2 nano-Se + Si, and T5 = fungicide (Score 250 EC at 1 mL/2 L). Same letter above the error bar indicates no significant differences.

**Table 1 microorganisms-11-00728-t001:** Details about the treatments. All treatments were conducted 4 times a season starting 30 days after sowing and repeated with 10 days intervals between each spraying (treated 30, 40, 50, and 60 days after sowing).

Code	Description of the Treatment (s)
T1	Untreated plants
T2	Foliar application of 100 ppm biologically synthesized nano-Se from Agricultural Microbiology Research Department (Giza, Egypt)
T3	Foliar application of 200 ppm fine-ground nano-Si from Agricultural Microbiology Research Department (Giza, Egypt)
T4	Foliar application of 50% of T2 and T3 (i.e., 50 ppm of nano-Se + 100 ppm of nano-SiO_2_)
T5	Spraying of commercial fungicide Score 250 EC (Difenoconazole, dose 1 mL/2 L from Syngenta (Basel, Switzerland)

**Table 2 microorganisms-11-00728-t002:** Disease severity percentage as the pathogenicity of 6 isolates of *Alternaria alternata* on common bean plants under greenhouse conditions, 21 days after inoculation.

Isolate No.	Disease Severity (%)
A1	41.30 b ± 2.05
A2	23.00 c ± 1.99
A3	27.15 c ± 2.00
A4	14.37 d ± 1.89
A5	78.23 a ± 3.47
A6	30.00 c ± 2.08
F. test	**

Where A1, A2, A3, A4, A5 and A6 are the six isolates of *Alternaria alternata*, which used in the study ** indicates highly significant treatment and values of means in each column followed by the same letter are not significantly at a 95% confidence level (±SD = standard deviation; N = 5).

**Table 3 microorganisms-11-00728-t003:** Effects of the different treatments on mycelial growth of *Alternaria alternata* and the corresponding reduction in the pathogen under in vitro conditions in the lab.

Treatments	Mycelial Growth (cm)	Reduction (%)
Negative control	9.0 a ± 0.21	0.001 f ± 0.00
Nano-Se (25 ppm)	7.0 b ± 0.17	22.22 e ± 0.95
Nano-Se (50 ppm)	4.5 c ± 0. 08	50.00 cd ± 2.47
Nano-Se (100 ppm)	1.7 d ± 0. 08	85.11 b ± 3. 11
Nano-Si (100 ppm)	6.0 b ± 0. 15	33.33 e ± 2.09
Nano-Si (200 ppm)	3.5 c ± 0. 11	61.11 c ± 3.01
1/2 Nano-(Se + Si) (i.e., 50 + 100 ppm, respectively)	2.0 d ± 0. 05	77.78 bc ± 3.34
Fungicide (Score 250 EC)	0.4 e ± 0. 01	95.55 a ± 4. 37
F. test	**	**

** indicates highly significant effect of treatment. Mean values in each column that are followed by the same letter are not significantly different from each other at a 95% confidence level (±SD = standard deviation; N = 5).

**Table 4 microorganisms-11-00728-t004:** Effects of applied nano-Se and nano-SiO_2_ treatments on disease severity and efficacy (%) of Alternaria leaf spot disease under field conditions at 45, 60, 75, and 90 days after the sowing of common beans during two seasons, where T1 = negative control, T2 = 100 ppm nano-Se, T3 = 200 ppm nano-SiO_2_, T4 = 1/2 nano-Se + SiO_2_, and T5 = fungicide (Score 250 EC at 1 mL/2 L).

Treatments	Disease Severity (%)				Efficacy (%)
	Days after Sowing Date				
	45	60	75	90	
**Season of 2021**					
T1	29.5 a ± 1.24	42.1 a ± 2.56	61.8 a ± 3.18	89.6 a ± 3.76	-------
T2	1.5 b ± 0.16	5.5 b ± 0.48	14.8 c ± 0.53	20.5 c ± 1.08	77.1 ab ± 2.55
T3	3.3 b ± 0.23	7.5 b ± 0.59	18.1 b ± 0.73	30.3 b ± 1.22	66.1 c ± 2.29
T4	0.0 b ± 0.00	5.8 b ± 0.49	15.1 c ± 0.48	23.5 bc ± 1.18	73.7 b ± 2.51
T5	0.0 b ± 0.00	6.2 b ± 0.61	12.5 d ± 0.67	17.8 c ± 0.79	80.1 a ± 3.05
F. test	**	**	**	**	**
**Season of 2022**					
T1	31.3 a ± 3.04	49.1 a ± 4.49	66.0 a ± 4.99	94.6 a ± 4.99	--------
T2	2.4 c ± 0.10	10.4 b ± 0.47	17.0 cd ± 1.02	23.0 c ± 1.06	75.7 a ± 4.04
T3	6.0 b ± 0.17	13.0 b ± 0.47	22.1 b ± 1.06	35.0 b ± 2.08	63.0 c ± 3.95
T4	3.5 c ± 0.14	11.7 b ± 0.47	19.1 c ± 1.06	27.8 b ± 2.05	70.5 b ± 4.04
T5	1.9 c ± 0.07	8.3 c ± 0.35	14.7 d ± 1.02	20.9 c ± 1.79	77.9 a ± 4.04
F. test	**	**	**	**	*

Mean values in each column that are followed by the same letter are not significantly different from each other at a 95% confidence level. * and ** indicate significant and highly significant, respectively; (±SD = standard deviation; N = 4).

**Table 5 microorganisms-11-00728-t005:** Vegetative growth traits, chlorophyll fluorescence, and chlorophyll content of common bean plants influenced by the treatments in the field experiments and where T1 = negative control, T2 = 100 ppm nano-Se, T3 = 200 ppm nano-SiO_2_, T4 = 1/2 nano-Se + SiO_2_, and T5 = fungicide (Score 250 EC at 1 mL/2 L).

Treatments	Stem Length (cm)	No. of Leaves/Plant	Plant Fresh Mass (g)	Plant Dry Mass (g)	Chlorophyll Fluorescence (F_V_/F_M_)	Chl. A	Chl. B
						(mg 100 g^−1^ FW)	
	**Season of 2021**						
T1	43.7 c	14.2 d	60.5 d	13.66 d	0.724 c	18.22 c	8.14 b
T2	54.0 a	19.1 a	101.5 a	22.14 a	0.798 a	23.99 a	9.23 a
T3	51.6 b	17.3 b	93.6 b	19.55 b	0.795 a	22.17 b	8.36 b
T4	53.3 a	17.5 b	95.2 b	20.05 b	0.795 a	23.05 ab	8.55 b
T5	50.6 b	15.7 c	84.6 c	17.99 c	0.771 b	23.99 a	9.18 a
F-test	**	**	**	**	**	**	*
	**Season of 2022**						
T1	40.4 c	11.8 c	57.1 d	12.45 d	0.733 c	17.09 c	8.02 a
T2	51.8 a	16.7 a	94.7 a	19.77 a	0.778 a	22.55 a	9.02 a
T3	48.9 b	14.7 b	86.3 b	17.56 b	0.776 a	20.08 b	8.34 a
T4	50.1 ab	15.5 ab	85.7 b	17.89 b	0.777 a	20.60 b	8.45 a
T5	48.5 b	14.5 b	79.4 c	16.05 c	0.754 b	21.95 a	8.88 a
F-test	*	*	**	**	*	**	NS

Values of means in each column followed by the same letter are not significant at *p* < 0.05 * or *p* < 0.01 **, respectively.

**Table 6 microorganisms-11-00728-t006:** Total dry yield of common bean seeds as influenced by different treatments. Results from the field experiments, and where T1 = negative control, T2 = 100 ppm nano-Se, T3 = 200 ppm nano-SiO_2_, T4 = 1/2 nano-Se + SiO_2_, and T5 = fungicide (Score 250 EC at 1 mL/2 L).

Treatments	Dry Seed Yield (g plant^−1^)	Dry Seed Yield (Mg ha^−1^)	Dry Seed Yield (g plant^−1^)	Dry Seed Yield (Mg ha^−1^)
	**Season of 2021**		**Season of 2022**	
T1	30.85 d ± 2.05	2.17 c ± 0.009	27.19 d ± 3.03	1.77 e ± 0.019
T2	54.22 a ± 2.79	3.07 a ± 0.017	45.75 a± 3.55	2.66 a ± 0.025
T3	38.99 c ± 2.48	2.48 b ± 0.025	32.27 c ± 3.35	2.24 d ± 0.033
T4	49.13 ab ± 2.75	2.77 a ± 0.032	40.18 b ± 3.44	2.49 b ± 0.029
T5	45.88 b ± 2.57	2.61 b ± 0.044	38.65 b ± 4.01	2.41 c ± 0.018
F-test	**	**	**	**

** indicates highly significant. Mean values in each column followed by the same letter are not significantly different from each other at a 95% confidence level (±SD = standard deviation; N = 4).

## Data Availability

Available on request.

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
