# Peer review of "Nanofungicides with Selenium and Silicon Can Boost the Growth and Yield of Common Bean (Phaseolus vulgaris L.) and Control Alternaria Leaf Spot Disease"

_microorganisms, 2023, doi:10.3390/microorganisms11030728_

Round 1

Reviewer 1 Report

The manuscript 'Nano-Fungicides with Selenium and Silicon can Boost the Growth, and Yield of Common Bean (Phaseolus vulgaris L.) and Control Alternaria Leaf Spot Disease' investigates nano selenium (nano-Se) and nano-silica (nano-SiO2) as alternatives against leaf spot disease. The well-written manuscript focuses on an interesting and topical subject and is of considerable practical importance; however, it is still need revised.

LINE109 What are the environmental conditions?

LINE 120-128 The scanning electron microscope (SEM) and UV analysis of nano-Se and nano-SiO2 are still need added in the attachment and marked with reference.

LINE 441 Alternaria, in italic type

LINE 512-526 Only the results of this study are retained, and other words can be explained in the discussion.

Author Response

Thank you for your comments. Please see the attachment for detailed feedback.

Reviewer 2 Report

It should be noted the relevance of the chosen topic.

From what culture were selenium nanoparticles synthesized?

In research, it would be good to determine the viability of plant cells by cytological methods.

The authors argue that the use of selenium increases the resistance of plants to phytopathogens due to the effect of selenium on the production of ROS. It would be nice to determine ROS in fungal stress.

It is necessary to bring the list of references in accordance with the rules of the journal.

Author Response

Thank you for your comments. Please the the attachment for detailed feedback.

Reviewer 3 Report

GENERAL REMARKS

This study reports the results of experiments conducted in vitro and in the field, in which nanoparticles preparations were applied to bean plants to test their efficacy against Alternaria-based diseases. The results showed that the treatments were effective in controlling the disease development and in stimulating plant enzymatic systems to the benefit of resistance capacity to infections.

Nanotechnology applied to plant pathology is an emergent research field, so this study is a welcomed contribution knowledge-building for this topic.

The manuscript is clearly written, although the English could still be improved, and the methodology appears appropriate and correctly applied.

I have some minor observations concerning some descriptive sections of the manuscript and the discussion, which could be made more concise, for the rest I do not object to the publication.

SPECIFIC REMARKS

Lines 192-206. The description of the enzymatic assays could be improved. It is evident that all these methods cannot be reported for brevity, and the references are enough.

Reporting methodological details out of a context unknown to the reader should be avoided, e.g., what is “mixture” (line 198)? And what is the “resulting solution”? If it is really necessary to describe methodological details (when, for example, significant modifications respect to the original procedure were made), describe them exhaustively, to made them comprehensible, otherwise omit them.

Lines 400 to 424 seem to me a mere repletion of concepts already expressed in the introduction. If you think that some more explanation is needed, just complete and refine the introduction, but all this section can be removed.

Lines 425 and 433 reports considerations that could be more appropriate in the materials and methods section, since they pertains to the whole rationale of the methodology.

Author Response

Thank you for your valuable comments. Please see the attached document for detailed feeback. We have carefully edited the manuscript and also made English grammer and language checks of the entire manuscript. Please note that all changes are marked with yellow in our revised manuscript.
